# CoMix: A Comprehensive Benchmark for Multi-Task Comic Understanding

**Emanuele Vivoli**[1,2]    **Marco Bertini**[2]    **Dimosthenis Karatzas**[1]

[1]Computer Vision Center, UAB, Spain    [2]MICC, University of Florence, Italy

{evivoli,dimos}@cvc.uab.cat

{emanuele.vivoli, marco.bertini}@unifi.it

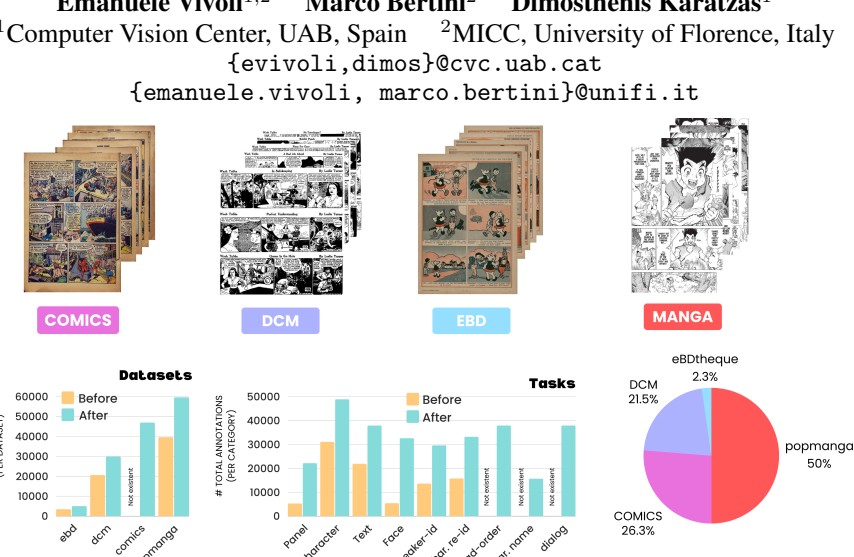

Figure 1: Composition of the *CoMix* benchmark. The **top** part of the figure provides a qualitative representation of the datasets included in *CoMix*. The accompanying bar charts depict the differences between the original annotations and those extended in *CoMix*. The **left** chart shows the increased number of annotations per dataset, whereas the **right** chart details the increase per task.

## Abstract

The comic domain is rapidly advancing with the development of single-page analysis and synthesis models. However, evaluation metrics and datasets lag behind, often limited to small-scale or single-style test sets. We introduce a novel benchmark, *CoMix*, designed to evaluate the multi-task capabilities of models in comic analysis. Unlike existing benchmarks that focus on isolated tasks such as object detection or text recognition, *CoMix* addresses a broader range of tasks including object detection, speaker identification, character re-identification, reading order, and multi-modal reasoning tasks like character naming and dialogue generation. Our benchmark comprises three existing datasets with expanded annotations to support multi-task evaluation. To mitigate the over-representation of manga-style data, we have incorporated a new dataset of carefully selected American comic-style books, thereby enriching the diversity of comic styles. *CoMix* is designed to assess pre-trained models in zero-shot and limited fine-tuning settings, probing their transfer capabilities across different comic styles and tasks. The validation split of the benchmark is publicly available for research purposes, and an evaluation server for the held-out test split is also provided. Comparative results between human performance and state-of-the-art models reveal a significant performance gap, highlighting substantial opportunities for advancements in comic understanding. The dataset, baseline models, and code are accessible at the repository link. This initiative sets a new standard for comprehensive comic analysis, providing the community with a common benchmark for evaluation on a large and varied set.

38th Conference on Neural Information Processing Systems (NeurIPS 2024) Track on Datasets and Benchmarks.

# 1 Introduction

Comics, a distinct medium that seamlessly blends textual and visual components, has emerged as a globally celebrated form of cultural expression. While the accessibility of comics allows even young readers to comprehend and appreciate them with ease, the intricacy of comic page layouts poses significant challenges for computational understanding. The classical elements of comics, including panels, speech balloons, characters, text, and onomatopoeia, are heavily shaped by the creator's imaginative vision and artistic flair, rendering the undertaking of comic image analysis a complex and multifaceted endeavor.

In recent works[33], authors approached comics research to more complex tasks moving from *classification*[32], *detection*[35] and *captioning*[34] to *diarization*[19, 28], where low-level tasks - such as detecting objects, defining reading order and speaker identification and character re-identification - are used as a medium to generate an ordered transcription of who said what on single page. Despite the innovative application of this complex bottom-up approach to comics understanding tasks, there is no benchmark on character naming or diarization, nor metrics to assess their correctness [33].

A number of available datasets support some of these tasks. For example, eBDtheque [10] and DCM772 [23] provide detection annotations for panels, characters, text lines, and some occurrences of character's faces. Another recent dataset called PopManga [28] offers annotations to character and text detection as well as speaker identification and re-identification. Lastly, the well-known Manga109 dataset [8] has seen various iterations that enriched the annotations landscape from just object detection and character re-identification to the late onomatopoeia detection and recognition [2] and speaker identification [19]. However, the most comprehensive dataset, Manga109, comprehends only manga-style Japanese comics, limiting a proper evaluation of generalization capability across different styles and lacking a diarization ground truth.

In this work, we propose the *CoMix* - a benchmark formed of purposefully selected and annotated comic books that aims to comprehensively assess the capability of single and multimodal models across different vision tasks (detection, re-identification, OCR), multimodal tasks (speaker identification, character naming, reading order, dialog generation), spanning different types of comics style (American, Manga and small percentage of French) and modalities (allowing for single page and multi-page). Our benchmark draws inspiration from datasets like Manga109 [8], and recent work like Magi [28], closing the gap between more capable models and datasets/metrics unavailability.

*CoMix* contains 3.8k images, gathered from 100 books, densely annotated with 130K objects across the four classes considered, 30k text-characters links, with 33k characters clusters with 16k total names identified. An overview of single-page annotations is provided in Figure 2 while statistics about datasets and annotations are provided in Table 2. We open-source the images and annotations of the validation splits. An evaluation server and images from the held-out test split are made available. Since currently there is no model that can tackle all the evaluation tasks in our benchmark, we provide baseline results for per-task models: object detection, speaker identification, character re-identification, reading order, character naming, and dialog generation.

The contributions of this work are as follows:

- We address the lack of metrics and benchmark datasets for comics understanding, proposing high-level tasks metrics: Hybrid Dialog Score for character naming and dialog generation;

- We introduce *CoMix*, a diverse manga- and comics-style benchmark of carefully selected books with computational and reasoning-dense annotations;

- We provide baseline results for each task, identifying substantial performance gaps with human baselines.

In the next section (section 2), we discuss related work in more detail, highlighting what sets the *CoMix* apart in the landscape of comics analysis datasets and benchmarks. In sections 3 and 4, we describe the books and annotations in the *CoMix*, with details about the diversity of comics and artistic styles. In section 5, we introduce the tasks enabled by these annotations, together with evaluation metrics and baselines, including a human baseline. We conclude with a summary and directions for future work in section 7.

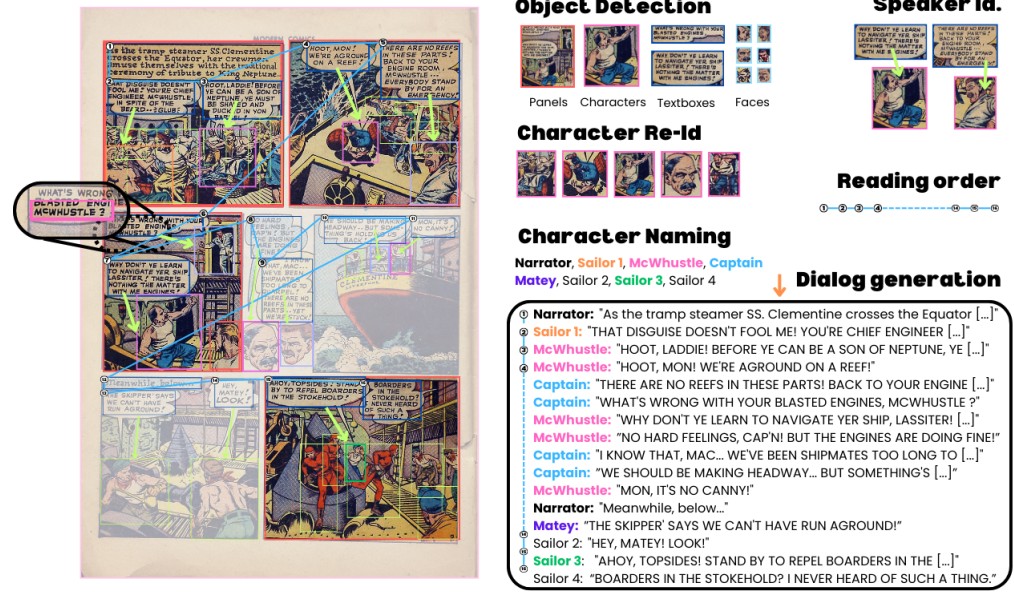

Figure 2: The *CoMix* benchmark contains 4 computational tasks (object detection, speaker identification, character re-identification, panel-text sorting) and 2 multi-modal reasoning tasks (character naming and dialog generation) which require models to detect objects and their relation, as well as reading text. The figure shows the annotations added for each comic page, and on the left is depicted an example annotation of multi-modal reasoning task *dialog generation*.

## 2   Related Work

A limited number of comics-related benchmarks exist in the literature, covering tasks such as classification (image classification [3, 5, 16, 30, 37], emotion recognition [24], action detection [15]), detection [1](panel, character, text, etc.), and modification (de-warping [9], image-to-image translation [31]), to cite a few. We focus the discussion here on detection and analysis benchmarks and highlight the differences between *CoMix* and prior work regarding the data collection process and available annotations and tasks.

One main limitation that affects all works related to comics (and art in general) is copyright issues. Among various datasets that have been proposed over the years, many no longer exist [22, 36, 14, 22], and many others are not available [17, 13, 29, 27, 6, 11, 4]. The majority of these datasets are designed for detection tasks, with images spanning from cartoon and sketches to French, American and Japanese comics. Only a few datasets are available whose annotations only assess classification [10, 15, 24] and detection [23, 8, 2].

Building on prior discussions, it's notable that existing datasets introduce sophisticated tasks such as speaker identification and re-identification; however, they predominantly feature manga-style comics and exhibit several critical shortcomings. Firstly, annotations are typically restricted to principal characters [19], limiting the scope of character detection and naming. Secondly, characters' names are often omitted [28], and thirdly, there are no established metrics or benchmarks for evaluating tasks like character naming and dialog generation. Furthermore, the Manga109 dataset [19] is exclusively composed of Japanese mangas, which presents challenges for global applicability. Therefore, there is a clear need for a more inclusive comic dataset that not only spans multiple styles but also provides dense annotations and comprehensive metrics for benchmarking.

Moreover, data sources are of huge importance. Sachdeva and Zisserman [28] selected the most popular manga of all time creating the PopManga dataset, gathering images from Manga Plus by Shueisha[1]. A similar approach was previously applied by Iyyer et al. [15], where a collection of 5k most-rated comic books was employed in constructing the COMICS, scraping images from the well-known Digital Comic Museum[2]. These approaches ensure that data is of high quality but it does

---

[1]https://mangaplus.shueisha.co.jp
[2]https://digitalcomicmuseum.com

Table 1: Characteristic of existing datasets (test split) compared to *CoMix*. Tasks: Classification (c), Detection (d), Text-Character association (t2c), Character Re-Identification (c2c), Character Naming (N), and Dialog (D). ✓ represent available datasets, while ✗ not, thus asterisks (*) represent reported numbers by authors. Highlighted rows are present in our benchmark dataset.

| Dataset | Release | Avail | Tasks | Years | Style | Books | Pages |
|---|---|---|---|---|---|---|---|
| eBDtheque [10] | 2013 | ✓ | d,t2c | 1905-2012 | mix | 28 | 100 |
| COMICS [15] | 2017 | ✓ | c | 1938-1954 | comics | 3948 | 198k |
| GCN [6] | 2017 | ✗ | d,t2c | 1978-2013 | comics | *253 | *38k |
| DCM772 [23] | 2018 | ✓ | d | 1938-1954 | comics | 27 | 772 |
| Manga109 [8, 25] | 2018 | ✓ | d,t2c,c2c | 1970-2010 | manga | 109 | 10k |
| BCBId [7] | 2022 | ✓ | - | - | bangla | 64 | 3k |
| VLRC | 2023 | ✗ | - | 1940-now | - | *376 | *7k |
| PopManga [28] | 2024 | ✓ | d,t2c,c2c | 2010-2023 | manga | 25 | 1.8k |
| *CoMix* (our) | 2024 | ✓ | d,t2c,c2c,N,D | 1938-2023 | mix | 100 | 3.8k |

limit the variability in style and complexity. In fact, most common manga reflect user preferences, which force drawers to standardized style. The same, together with high-quality scans, happen to appear in comics. Moreover, as the sampled books correspond to the same collections, characters and styles for the PopManga appear to be the same, as in fact the PopManga unseen split is the collection of almost 1k images, but only spanning 10 different sagas.

Table 1 summarises the characteristics of the *CoMix* compared to previous efforts. It can be observed that the *CoMix* has better coverage of annotations. We emphasize that the *CoMix* is not designed to be a large-scale dataset. Instead, it is an evaluation benchmark, with limited but densely annotated data, meant to assess the multi-task capabilities of models.

## 3 Books in the *CoMix*

The *CoMix* dataset has been meticulously curated to showcase a broad spectrum of comic book styles, drawing samples from Japanese manga (PopManga), American comics (DCM and newly collected Comics), and French Bandes Dessinées (eBDtheque). Although multilingualism is not a primary objective, some pages from eBDtheque include French and Japanese, reflecting incidental multilingual aspects.

**Comics choice:** As depicted in Table 1, the integration of existing datasets—PopManga, DCM, and eBDtheque—reveals a predominant bias towards manga-style comics. To address this imbalance, we strategically augmented the dataset with a selection of American comics from the Digital Comic Museum, which features over $22k$ golden-age American comic books. Popular characters such as *"Plastic Man"* and *"Daredevil"*, are drawn from the most downloaded comics, although selection criteria were refined beyond download counts due to potential skew from non-representative downloads [3]. Crucially, each book's metadata links to the Grand Comics Database [4], providing detailed character and storyline annotations that facilitated the selection process.

Using this information, we selected comic books based on the distribution of character appearances across the books following the principle of (i) most characters possible, from the minimum number of books, and (ii) all possible instances of these characters should be in the 100 books selected, of which 20 goes to the test/val splits. In constructing *CoMix*, our primary goal is to underscore books featuring characters that recurrently appear across various publications, indicating significant narrative roles. To achieve this, the selection method computes the ratio of shared-to-unique character appearances for each book, thereby ranking and choosing the top books based on this ratio. Specifically, this ratio assesses the frequency of characters' appearances within the same book against their appearances in other books. To further refine our selection, the algorithm was designed to maximize the diversity of characters across the selected books, prioritizing the retention of books that uniquely feature

---

[3] See comments in Wanted Comics 11 -JVJ

[4] https://www.comics.org/

Table 2: Annotations in the *CoMix* benchmark. The reported numbers correspond to both test (80 books) and validation (20 books) splits. Each detected object is represented by the class and bounding box. Text elements also possess ground truth transcriptions. Character elements, whenever relevant, are associated with the name. Not all texts are associated with a character, but all the texts have Reading order ground truth.

| Annotation type | # anns |
|---|---|
| Object detection (4 classes) | 130k |
| Speaker identification | 29k |
| Character Re-Id | 59k |
| Reading order | 3.9k |
| Character Naming | 15k |
| Dialog generation | 3.9k |

these characters. This ensures a broad representation of characters, with most being exclusive to the selected subset and not featured in the broader set of $22k$ comics. The ratio is then exponentiated (in our implementation, squared) to enhance distinctions between books with high and low levels of character sharing. After calculating these ratios for all books, they are ranked in descending order based on their scores, leading to the selection of the top 100 books. This metric, referred to as the "Pow Selection Approach", is detailed in the Supplementary materials. This approach ensures that the selected books reflect broader narrative arcs and character interconnectivity, thereby enriching the *CoMix* dataset's utility for diverse comic book analysis.

**Splits:** The *CoMix* contains 80 books from different sources, each of 44 pages on average, with a total of 3.5k images. This corresponds only to the held-out test split available through the evaluation server. Moreover, a validation split (20 books, 466 images), on which we tested and reported the results in this paper is provided.

# 4   Annotation in the *CoMix* benchmark

We annotate these comics with six types of annotations to cover low-level and high-level aspects, both computational and reasoning tasks. We enable various evaluations: object detection, speaker identification, character re-identification, character naming, reading order, and dialog generation. We include a summary of the number of annotations in Table 2 and visualizations in Figure 2.

**Object detection.**   Object detection represents the root annotation of our benchmark. All the other annotations, except for dialog generation, are linked or grounded into detected objects. In the annotation process, we instructed annotators to focus on all size elements of the four selected categories: panels, characters, faces, and text boxes. Regarding other possible comic-related classes, we decided not to consider balloons and onomatopoeias, nor scene text not relevant to spoken text. Many of these classes were already annotated in the existing datasets, but with different design choices: PopManga does not have faces and panels, DCM does not have faces, and none of them have subsequent complex annotations that we describe next. In the supplementary material, we show our annotations compared to existing ones. Regarding characters, following [10], we consider human-like, animal-like, and object-like characters to be annotated, without explicitly differentiating these. When the characters are partially occluded, the annotators mark only the visible part of the character's body as boxes. Moreover, when the character is not recognizable if not thanks to in-page context (i.e. in a large zoom-out scene), we ask to annotate the so-called character. Some ambiguous character detection remains, like multi-page zoomout, or back-and-forth change of scene among pages. A list of specific examples is included in the supplementary materials. Finally, we consider faces where the nose is visible, independently from the face-side (front, side, or backward).

**Speaker identification.** Although object detection allows for some initial comics analysis, such as global comics layout with panels, character pose variability, and the usual aesthetic and location of text and dialogs, they do not fully describe a comic page. A better understanding of compositional interaction with objects arises by linking the text boxes (spoken texts) to the respective speaking characters. To this end, following [19], we annotate oriented polylines (from text to character) on the

Table 3: Computational tasks, and top-performing baselines in the *CoMix*. The dialog proposed metric "Hybrid Dialog Score" is indicated with HDS.

| Task | Output | Metric | Baseline | Score |
|---|---|---|---|---|
| Object detection | box detection | mAP - $R$@100 | Magi | 78.6 - 67.9 |
| Speaker identification | object indexes | $R$@#text | heuristic | 0.68 |
| Character Re-Id | cluster ids | AMI - NMI | DINOv2 | 0.29 - 0.51 |
| Character Naming | names | ANLS | GPT-4 | 47.11 |
| Dialog generation | list of tuples | HDS | GPT-4 | 93.14 |

respective objects. Annotators were instructed to select two points, text, and characters, inside the respective boxes. These points are post-processed to obtain (text, character) pair indexes.

**Character Re-Identification.** Another linking task comprehends re-identify characters within the same page. This task can be seen as binary classification among all the possible pairs of characters (as approached in [28]) or as a clustering task. When the characters are difficult to recall (i.e. recognizable only by looking at the context, even when reading the panels) we still ask the annotators to put in the effort and give the right identity to the character. These cases appear many times, especially in comics-style, as characters are often not consistent in little details (see example in the supplementary materials for a better overview).

**Reading order.** Mangas and Comics are read differently. Mangas from right to left, while Comics from left to right. Both are read top-down. Despite the good performances of cut-based panel sorting algorithms [18, 12] and graph-based approach [28], we decided to extensively annotate also reading order to assess the corner-cases of these so-common used euristics.

**Character Naming.** Differently from the existing datasets [8], which assign a name tag only to main characters, we extensively annotate names assigning to non-primary characters an exhaustive description of their role (i.e. "Captain", "Sailor") which ensure a non-zero metric score in character naming metric (see Section 5 for an overview of the metric). However, not every character can have a specific description, thus, we rely on incremental indexes to name different characters with the same roles (i.e. "US soldier 1", "US soldier 2", etc.). Whenever the character names are spoken in the text boxes, their names propagate on the whole story.

**Dialog Generation.** As a way to connect high-level and low-level single-page comics understanding capabilities, we define dialogs as an ordered list of tuples with the speaker name (character names or "narrator"). The texts are sorted based on the ground truth reading order. Annotators are instructed to maintain punctuations and text format (lower-case and upper-case) but discard other text properties (bold, italics, canceled, or underlined text).

# 5 Baselines results

**Computational tasks:** We defined six computational tasks based on the annotations available in the *CoMix* dataset. We summarise in Table 3 the task definitions (outputs, metrics) and the performance of top-performing baselines. It can be observed that the *CoMix* combines lower-level dense prediction tasks like object, speaker identification, and character re-identification, whose outputs are box and group of indexes, with higher-level tasks like dialog generation. More details about the task definitions are included in the supplementary materials.

**Baselines:** Ideally, a single model should be able to perform all the tasks in the *CoMix* benchmark. Since such a model is not available in the literature, we include results obtained with per-task baselines on the validation split for all the six tasks in the *CoMix*; see Table 3 for a summary of top-performing baselines and their average performance, and the supplementary materials for more details. When selecting and running these baselines, we favored the same approaches used by Sachdeva and Zisserman [28], which we carefully detail in the following section. However, for character naming, such models do not exist in the literature, so we evaluated namings together with dialogs, instead of with detection and clustering.

In the following section, we provide, for every benchmark, an explanation of the used metrics and an overview of the methods employed in the benchmark.

**Object detection:** For object detection, we selected two common metrics: mean Average Precision at IoU of 0.5 ($mAP@0.5$) and Recall at 100 objects ($R@100$). We benchmarked a variety of models including convolutional and transformer-based architectures, with a focus on their performance in both fine-tuned and zero-shot settings on comic data. Among these, GroundingDino [20], designed for open-set object detection using natural language object classes, was a key zero-shot model utilized to detect four classes: panels, characters, text, and faces. Additionally, conventional models such as Faster R-CNN, SSD, and YOLO were trained on varied comic styles to assess the impact of different training data distributions. For characters and faces detection we employed also a YOLOX-based available model named DASS [31]. Notably, Magi [28], a transformer-based model, was included for its impressive capability demonstrated in Manga-style comics. Details about the fine-tuned training procedure and GroundingDino prompts for zero-shot detection are provided in the appendix, as well as more detailed benchmark results.

**Text-Character association.** The task of associating a speaking character to a textbox is not new, and neither is the metric. Following previous work [19], we employ classical Recall@K, with the value of K indicating the number of textboxes on a page, called $R@\#text$. The global score is obtained by averaging a single-page score. The benchmarks are composed by: (i) an existing heuristic approach connecting the textbox to the closest character [23]; and (ii) Magi, which is trained for providing these associations among the detected textboxes and characters. In this case, differently from what is shown by Sachdeva and Zisserman [28], the heuristic of "most close character" works better. We speculate this drop in performance is given by the style shift between manga (on which Magi has been trained) and comics.

**Character Re-Identification.** For the task of Re-Identification (also known as single-page character Clustering), we employed Adjusted Mutual Information ($AMI$) and Normalized Mutual Information ($NMI$) to evaluate performance. $AMI$ measures the agreement between the clustering results and the ground truth labels, adjusted for chance. $NMI$, on the other hand, normalizes mutual information by the potential disorder in each set of labels, thus reflecting the purity of the clustering. Contrary to previous other studies, we did not utilize retrieval metrics such as $MRR$, $MAP$, or $Precision@k$. These metrics are heavily dependent on the presence of relevant items within the retrieved sets. Given that our analysis involves clustering where unequivocally relevant retrievals are absent — often resulting in clusters of a single element — these metrics would inherently score zero, thus failing to provide meaningful insights into our clustering approach's effectiveness. We benchmarked CLIP and DINOv2 as feature extraction models calculating the best clusters at max $AMI$ score, following [28]. Differently from what was previously reported, DINOv2 obtained higher scores than the fine-tuned model competitor Magi, thus indicating that despite maintaining high detection scores, Magi is not able to retain recognition performances out of its manga domain.

**Reading order.** For the task of reading order, we propose a simple edit distance metric on the sorted detected textboxes matched with the ground truth. As Magi performances in detecting textboxes and panels retain high accuracy, as benchmark adapted the Magi algorithm to operate both for manga and comics style. Prior knowledge makes it possible to know that comics pages are read from top to bottom, with an orientation difference between manga (right to left) and comics (left to right). Once the detected panels are sorted (with DAG approaches described in [28], adapted for comics), the textboxes within the panels are ordered based on the vicinity with the panel's top-right corner (manga) or top-left corner (comics). The result is reported in Table 3, and an overview of the panel DAG is given in the supplementary materials.

**Character Naming & Dialog Generation.** In addressing the challenges of character naming and dialog generation, we acknowledge that there is a lack of specific metrics. In previous works, the dialog is generated with a combination of algorithmic panel ordering, character progressive naming ("character 1", "character 2"), and fine-tuned OCR for textbox transcription. This engineering and multi-step approach is a good showcase but is not evaluated with some metric against any ground truth. We introduce a smoothed case-sensitive edit distance metric called "Hybrid Dialog Score" ($HDS$) to evaluate the accuracy of model outputs against ground truth dialog annotations. This metric assesses both the precision of transcribed dialogues and the accuracy of character identification in a unified framework. The metric operates in three steps: (i) we match ground truth and predictions texts using Hungarian matching with edit distance metric; for every match, we calculate (ii) the textbox edit

distance (normalized on ground truth text length) and (iii) the character name ANLS. An in-depth overview of the metric is provided in the supplementary materials, together with pseudocode. For our benchmarks, we utilized Magi as described in [28]. As MAGi is not able to detect names, we employ GPT-4 [26], leveraging its multimodal capabilities to interpret comic pages and generate structured outputs that include both dialogues and character names. GPT-4 was tasked with identifying and naming characters, where known, or assigning incremental identifiers otherwise, and transcribing dialogues exactly as they appear in comic format, respecting the sequence and case sensitivity. This approach not only captures the complexity of comic narratives but also enhances the evaluation of dialog transcription fidelity and character consistency across various comic styles. The result is reported in Table 3, and an overview of the GPT-4 prompt to obtain structured predictions is given in supplementary materials.

# 6 Ethical Considerations

**Copyrights and Consent.**   Comics, as a form of artistic expression, are governed by copyright laws that restrict access and usage. Our dataset, *CoMix*, aggregates comics from diverse sources: American comics from the Digital Comic Museum, manga from PopManga, and French Bande Dessinée from eBDtheque. In particular, *Digital Comic Museum* contains public domain assets, either released without copyright or with expired rights, allowing free research use [21]. The *PopManga* images are publicly accessible on "Manga Plus by Shueisha" with official permissions from copyright holders. The *eBDtheque* provides publicly available data cleared for non-commercial research use.

To ensure compliance, we have reorganized the dataset's structure, enabling users to acquire images directly from original sources and utilize our repository tools for formatting and validation [35]. This approach maintains adherence to copyright norms while promoting dataset accessibility and replicability.

**Data Quality and Representativeness.**   The *CoMix* benchmark evaluates models across various comic styles—American, Japanese, and European—to represent major comic production hubs. Predominantly comprising out-of-copyright works, especially American comics from the 1950s, the dataset may inherently reflect the social biases and stereotypes of that era. To address potential biases, we ensure a balanced representation of comic styles and origins to minimize cultural bias (Diverse Dataset Composition) and incorporate statistical analysis of factors like gender, ethnicity, and language representation (Bias Detection and Analysis).

On this regard, our first analysis concerns biases related to appearance (color vs. black-and-white) and types of characters in the comics. The color statistics are straightforward: by examining the first pages, we could determine whether each comic chapter is in color or black-and-white, as shown in Table 4a. For character type analysis, we classified character crops into one of four categories listed in Table 4b, using a state-of-the-art open-source MLLM[5]), which is known for its strong visual recognition and instruction following capabilities.

While the dataset shows a bias towards male characters, this is considered a reflection of actual character type distributions in real comics and manga, rather than a flaw in the dataset's quality. This dataset is intended for tasks such as dialog transcription testing.

Table 4: Initial Statistics in the *CoMix* Dataset

(a) Color vs. black-and-white images.

| Type | Percentage |
|---|---|
| Color | 59.2% |
| Black-and-White | 40.8% |

(b) Character Types in the *CoMix* Dataset

| Character Type | Percentage |
|---|---|
| Male | 74.3% |
| Female | 17.9% |
| Animals | 6.3% |
| Other | 1.5% |

---

[5]MiniCPM-llama3-v-2.5 at `https://huggingface.co/openbmb/MiniCPM-Llama3-V-2_5`

Future iterations will expand to include underrepresented styles such as Webtoons and Manhwa, alongside additional languages, to enhance global applicability and reduce cultural over-representation. Detailed statistics and bias analyses are provided in the supplementary materials.

**Semantic Harmful Content.** Our dataset has been meticulously curated to exclude NSFW and offensive content. All included works comply with the Comics Code Authority guidelines, ensuring appropriateness for diverse research and educational purposes. We conducted an automatic analysis of semantic content using the Llama3-80B model to classify panel-level text for offensiveness. Initial findings indicate low levels of harmful content (less than 0.5%), with detailed results and model prompts available in the supplementary materials.

Overall, these ethical considerations ensure that our research adheres to high standards of integrity, fairness, and respect for diverse cultural norms [21].

# 7   Conclusion

We introduce *CoMix*, a novel benchmark for multi-task and multi-modal comic analysis that addresses the limitations of existing datasets by incorporating diverse comic styles—including American, manga, and French—and providing comprehensive annotations across a wide range of tasks. *CoMix* encompasses fundamental vision tasks such as object detection and character re-identification, alongside complex multi-modal reasoning tasks like character naming and dialogue generation. The introduction of the Hybrid Dialog Score offers innovative metrics for evaluating these advanced tasks.

Baseline evaluations reveal significant performance gaps between state-of-the-art models and human performance, highlighting the inherent challenges in achieving nuanced understanding of the interplay between visual and textual elements in comics. By releasing the *CoMix* validation split and establishing an evaluation server for the held-out test split, we promote open research and facilitate robust benchmarking. *CoMix* sets a new standard for comprehensive comic analysis, providing a diverse and challenging testbed that will drive the development of more sophisticated and generalizable models capable of human-like comprehension in this culturally rich medium.

## Acknowledgments and Disclosure of Funding

This paper has been supported by the Consolidated Research Group 2021 SGR 01559 from the Research and University Department of the Catalan Government, and by project PID2023-146426NB-100 funded by MCIU/AEI/10.13039/501100011033 and FSE+. This work has been also funded by the European Lighthouse on Safe and Secure AI (ELSA) from the European Union's Horizon Europe program under grant agreement No 101070617.

Moreover, we are grateful to Digital Comics Museum for providing a free accessible source of comic books and to Grand Comics Database for the effort of building and maintaining correct and updated versions of comic memories. We want to thank Mohammed Ali Souibgui and Marco Mistretta for providing insightful input and Niccolò Biondi, Irene Campaioli, and Mariateresa Nardoni for being part of the revision and annotation team.

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
