# CoMix: A Comprehensive Benchmark for Multi-Task Comic Understanding

## Supplementary Materials

This supplementary document complements the main paper by providing additional information and examples that could not be included within the page constraints of the original manuscript. The structure of this document is as follows: in Section 1 details about the models and their usage. In Section 2 we presents an overview of the annotations from existing datasets and compares them with those in the *CoMix* benchmark. In Section 3 we detail the selection process of American Comics books included in our study. Extensive results are provided in Section 4, including those reported in the main manuscript. Finally, copyright and biases information is outlined in the Ethical Section 5. This supplementary material is intended to enhance the understanding and transparency of the research presented in the main paper.

## 1  Detailed Model Descriptions and Settings

**GroundingDino:** For zero-shot detection, GroundingDino was pivotal, using an array of class prompts to adapt to comic-specific elements. The class prompts used for detection were:

- **Panels**: "comics panels", "manga panels", "frames", "windows"
- **Characters**: "characters", "comics characters", "person", "girl", "woman", "man", "animal"
- **Text**: "text box", "text", "handwriting"
- **Faces**: "face", "character face", "animal face", "head", "face with nose and mouth", "person's face"

These prompts enabled the model to flexibly identify and classify a wide range of comic book elements by interpreting each class through the lens of natural language descriptions.

**DASS:** A convolution-based model utilizing the YOLOX architecture, DASS was developed in a self-supervised setup using a distillation approach from a teacher network with OHEM loss. It includes three variants—DCM, manga109, and mix—each fine-tuned on the dataset reflecting its name, optimized for detecting styles consistent with its training data.

**Standard Models:** Faster R-CNN, SSD, and YOLO models were adapted for comics by training them on comics, manga, and mixed datasets. These models were initialized with standard configurations and then fine-tuned to tailor to comic data, with adjustments such as changing the output classes to four and modifying learning rates and decay settings to optimize performance. Specifically, Faster R-CNN was adapted using a ResNet-50 backbone and trained with a learning rate of $5e^{-3}$, along with employing both StepLR and CosineAnnealingLR schedulers to manage learning rate adjustments across epochs. YOLOv8 and SSD have been trained using default configurations from "ultralytics" [1] and "mmdetection"[2] frameworks, respectively.

**Magi:** The transformer-based Magi model, following RelationFormer architecture [6], integrates a DeTr backbone with two MLP heads, focused on speaker identification and character re-identification. Initially pre-trained on a noisy dataset from Mangadex annotated with GroundingDino and later fine-tuned on a specialized popmanga dev-set, Magi exemplifies advanced model training with a focus on specific comic interactions.

---

[1]https://github.com/ultralytics/ultralytics
[2]https://github.com/open-mmlab/mmdetection

## 2 Annotations overview

This section offers an overview of the annotation differences in the *CoMix* benchmark compared to previous standards. Specifically, distinctions are highlighted in Figures 1, 2, and 3, where the "before" annotations are displayed on the left and the "after" on the right.

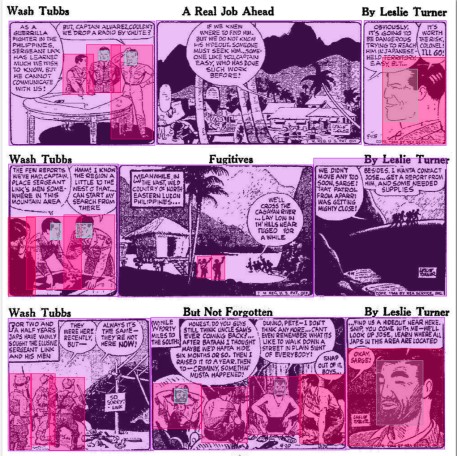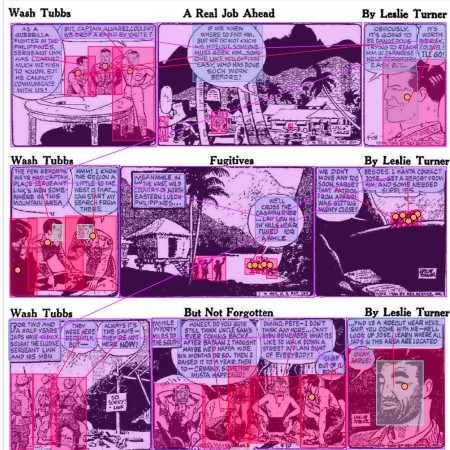

Figure 1: Image from "DCM", original annotations (left) and our *CoMix* corrected and integrated annotations (right). Every point indicates a re-identified character.

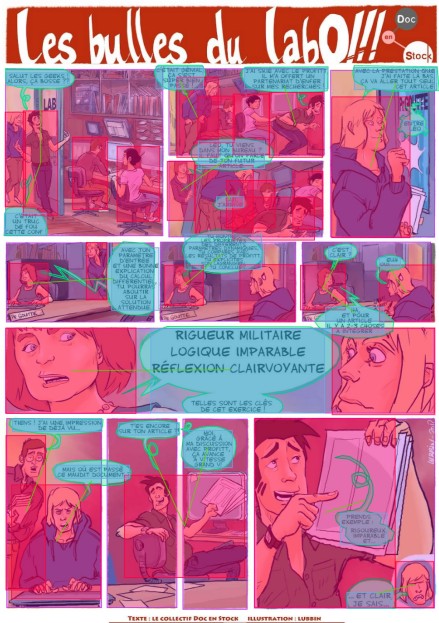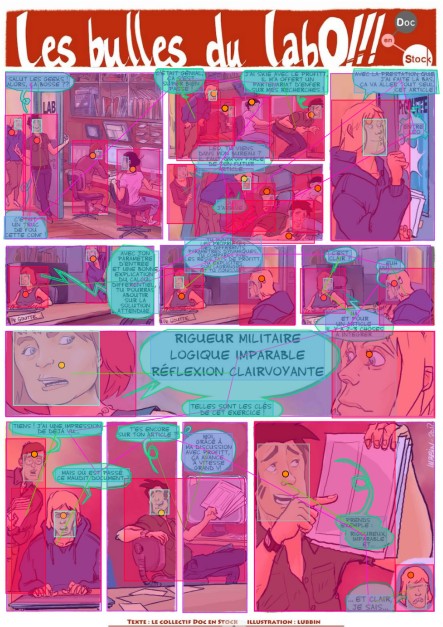

Figure 2: Image from "eBDtheque", original annotations (left) and our *CoMix* corrected and integrated annotations (right).

We state in the main manuscript that our benchmark does not include detection of objects such as "Balloons", "Onomatopoeias", and "scene text" (non-spoken text boxes). This decision is based on two main considerations:

**(i) Balloons:** Often, the annotated textboxes (spoken texts) are located inside balloons, making balloons essentially wrappers for the text. However, sometimes the text appears in a narrative box without a contour, making balloon detection a weak approximation.

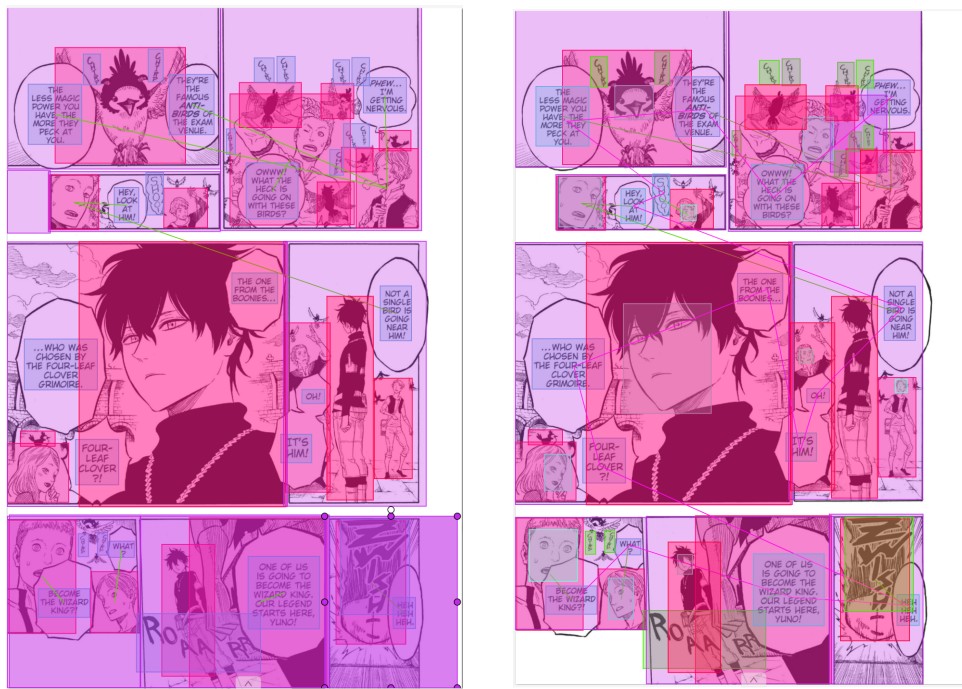

Figure 3: Image from "PopManga", original annotations (left) and our *CoMix* corrected and integrated annotations (right).

**(ii) Onomatopoeias and Scene Text:** If the text is spoken by a character, it is detected as "textbox", different from other approaches such as [1]. When onomatopoeias illustrate sound effects, we omit them to focus exclusively on spoken text, aligning with [3]. Unlike [5], we do not annotate scene-text to ensure models focus on crucial text (spoken) and avoid distractions from the scene.

Despite eBDtheque offering balloon annotations as seen in Figure 2, we chose not to eliminate these annotations. Similarly, PopManga, shown in Figure 3, provides bounding boxes for all text present in the scene, including spoken text, onomatopoeias, or scene text. We differentiate these by introducing onomatopoeia and scene-text classes without utilizing them in our analyses. These annotations are preserved for potential future research.

Additionally, annotations for speaker identification, character re-identification, and reading order may be less visible in the images provided. These are represented as green and red lines connecting the text to the character boxes (speaker identification), colored points identifying each character bounding box (character re-identification), and purple splines connecting the text boxes in reading order orientation (reading order). For a clearer view of these annotations, we will release the validation set images and annotations on the project website[3].

High-level annotations such as character naming and dialogue generation are not shown in these images but are part of our detailed annotation framework.

## 2.1 Quantitative details

In Table 1 and Table 2 are provided the original annotations numbers and the one we provide in *CoMix*, comparing images from the same sources. The first row, of every annotation category, provides the existing annotation number, while the second row is the one in *CoMix*. A third row provides the difference in percentage. From Table 1 we can notice that almost all the annotations categories experienced a substantial increase (with some also 100%) apart from the Panel detection in eBDtheque that we fixed eliminating duplicates and redundant panels boxes. For Table 2, almost all annotations in *CoMix* were not present before.

---

[3]Repository link: `https://github.com/emanuelevivoli/CoMix-dataset`.

Table 1: Summary of the Detection annotations across different datasets

| Category | Data Type | | | |
|---|---|---|---|---|
| | **DCM** | **EBD** | **Comics** | **Pop** |
| **Panel** | 4.5k
4.6k
+2.22% | 0.85k
0.84k
-1.18% | -
6.7k
+100% | -
9.9k
+100% |
| **Character** | 10.8k
11.4k
+5.56% | 1.6k
2k
+25.00% | -
15.9k
+100% | 18.8k
19.5k
+3.72% |
| **Text** | -
8.4k
+100% | 1.1k
1.1k
equal | -
11.9k
+100% | 20.8k
16.5k
-20.67% |
| **Face** | 5.4k
5.5k
+1.85% | -
1.1k
+100% | -
12.5k
+100% | -
13.6k
+100% |

Table 2: Summary of Higher-Level annotations across different datasets

| Category | Data Type | | | |
|---|---|---|---|---|
| | **DCM** | **EBD** | **Comics** | **Pop** |
| **Speaker ID** | -
6.2k
+100% | -
0.9k
+100% | -
8.9k
+100% | 13.6k
13.7k
+0.74% |
| **Character Re-ID** | -
7.4k
+100% | -
1.5k
+100% | -
8.5k
+100% | 15.8k
15.8k
equal |
| **Reading Order** | -
8.4k
+100% | -
1.1k
+100% | -
11.9k
+100% | -
16.5k
+100% |
| **Character Naming** | -
4.4k
+100% | -
0.5k
+100% | -
6k
+100% | -
4.7k
+100% |
| **Dialogue Gen.** | -
8.4k
+100% | -
1.1k
+100% | -
11.9k
+100% | -
16.5k
+100% |

## 2.2 Qualitative details

A notable example illustrating the design choices in the *CoMix* annotations is found in the densely populated pages such as the "Naruto" page from PopManga (Figure 5). This page features over 50 instances of the character Naruto, replicating himself using the "Shadow Clone Technique"—a concept well-known among the fan base. However, in PopManga, such a page receives only partial annotations, primarily highlighting large-scale depictions and main characters. This selective approach underscores our focus on significant elements over exhaustive detailing, which aligns with our annotation strategy to emphasize clarity and relevance in highly complex scenes.

## 3 Data selection

The *CoMix* benchmark sees the presence of selected American golden-age comics from DCM [4]. These documents have not been selected following the "most downloaded" principle, as instead is done in COMICS [2]. This, as mentioned in the paper, is because the DCM website reports gross download numbers, not caring about single account downloads. This means, as reported in one comment on the website[5], that repeatedly downloads by automatic bots count as single downloads, thus invalidating the global score validity. Instead, we want to guarantee the presence of various style comics, featuring as many characters as possible and whose characters are mostly present in these selected books. Thus we propose an algorithm to select the books with the required principles: "Pow Selection Approach", reported in the Algorithm 1.

### 3.1 Books selection algorithm

The book selection process emphasizes books containing characters frequently appearing across multiple books, suggesting greater narrative importance. This involves two key phases:

1. **Calculation of Shared-to-Unique Character Ratio:** For each book, a ratio is calculated based on the frequency of characters appearing within and in other books. To emphasize differences between books regarding character sharing, we squared the ratio.

2. **Selection of Top Books:** Books are ranked by their calculated ratios and the top 100 books with the highest scores are selected for further analysis.

---

[4]Digital Comics Museum at https://digitalcomicmuseum.com
[5]Comment on the most downloaded book Wanted Comics 11 -JVJ

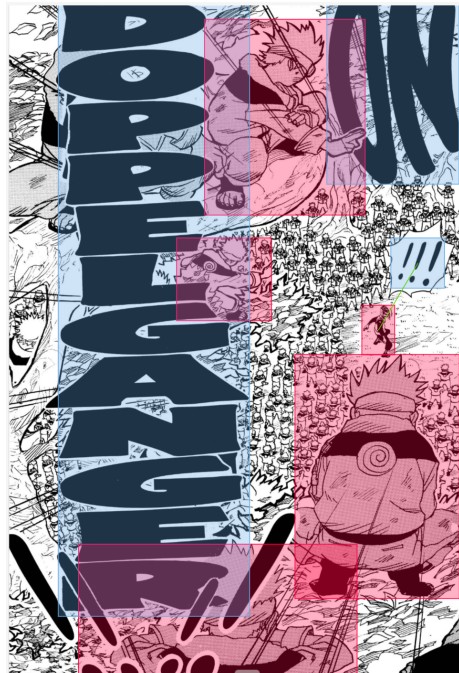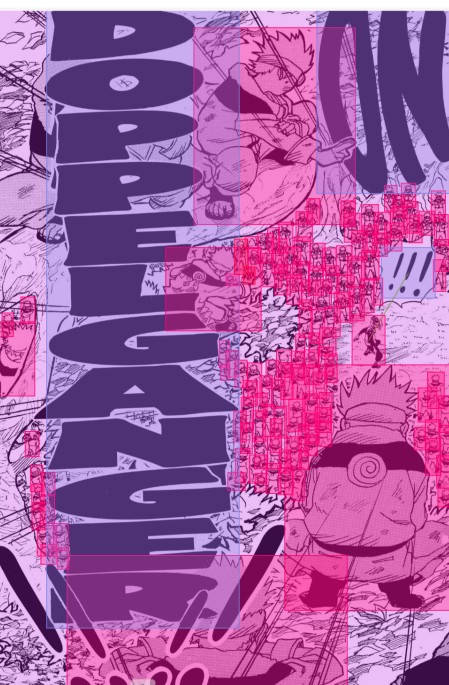

Figure 4: Image from "Naruto" (the "Shadow Clone Technique" from Chapter 2), annotated in the original PopManga dataset (left) and our *CoMix* benchmark (right).

The pseudocode is provided in Algorithm 1. This algorithm ensures that the selected books reflect a broader narrative context, as it prioritizes those with characters that bridge multiple storylines. Such an approach is suitable for analyzing complex datasets where character interrelations are significant.

---

**Algorithm 1** Book Selection Based on Character Sharing

---

1: **procedure** POWSELECTIONAPPROACH( *book_to_characters*, pow)
2:      *book_shared_to_unique_ratio* ← empty dictionary
3:      **for** *book_id*, *characters* in *book_to_characters* **do**
4:          *shared_count* ← 0
5:          *unique_character_count* ← len( *characters* )
6:          **for** *other_book_chars* in *book_to_characters.values* **do**
7:              **if** *other_book_chars* ≠ *characters* **then**
8:                  **for** *char_id* in *characters* **do**
9:                      **if** *char_id* in *other_book_chars* **then**
10:                         *shared_count* ← *shared_count* + 1
11:                    **end if**
12:                **end for**
13:              **end if**
14:          **end for**
15:          *ratio* ← $\frac{shared\_count}{unique\_character\_count}$
16:          *book_shared_to_unique_ratio*[*book_id*] ← *ratio*$^{pow}$
17:      **end for**
18:      **return** *book_shared_to_unique_ratio*
19: **end procedure**

---

### 3.2 Comics books overview

In this section, we provide details on the featured characters across the books and the selected ones. In Figure 6, on the left, a heatmap representing the occurrences of characters (y-axis) across different

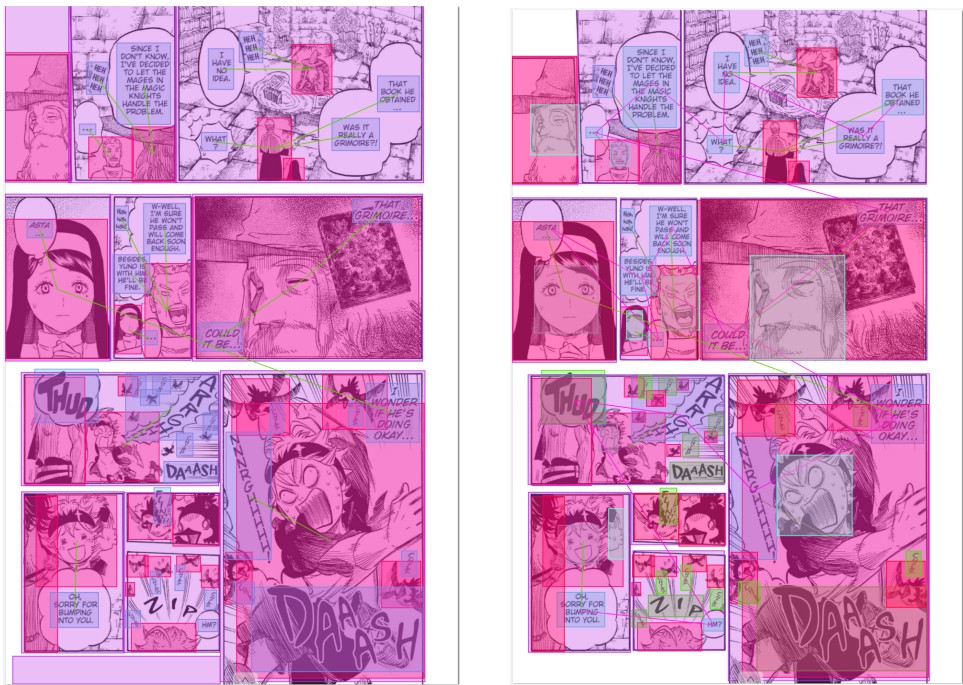

Figure 5: Image from "Black Clover" from the first chapters, annotated in the original PopManga dataset (left) and our *CoMix* benchmark (right). In the *CoMix* there are cleaned panel annotations, new faces boxes, corrected text to onomatopoeias annotations to and but also

books (x-axis) considering the number of pages the character appears in. The number of pages is given by

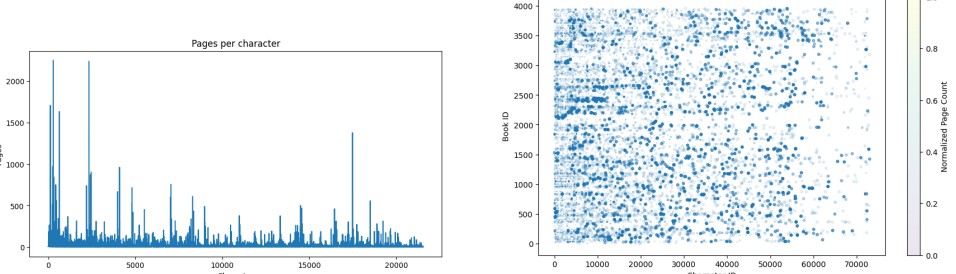

Figure 6: Overview of the characters per books. The bar-plot (left) represents the number of pages the characters appears in total. The right one, an overview of the presence of every character across books. Both graphs are generated using a first filtered subgroup of 4k books across the total 22k from DCM.

An additional step we have employed corresponds to filtering books that only contain one character and characters that are present in only one book (thus, the characters that have the spiking bar plot in Figure 6). We end up with a reasonable number of books (2k) for which more than 4k characters are present. 7. We calculated the Algorithm 1 on this collection of books.

# 4 Detailed Results

## 4.1 Detection

In particular, for the detection baselines, we have fine-tuned two convolutional-based architectures, Faster R-CNN and YOLOv8, previously employed in Comics Object Detection, and utilize the

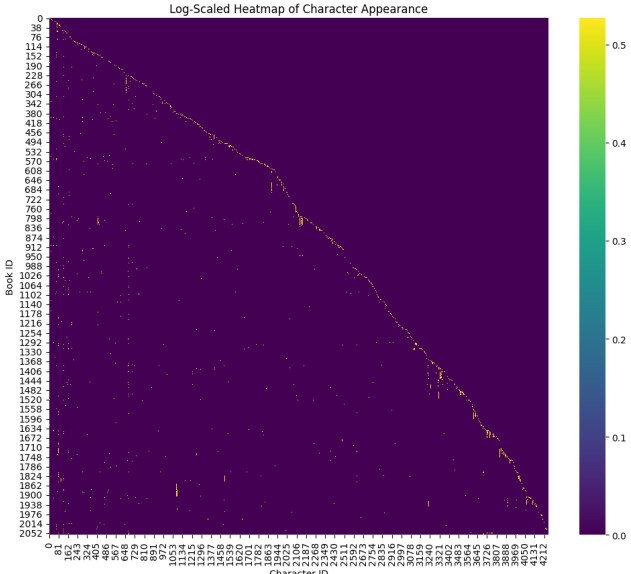

Figure 7: Overview of the characters per book, after the filtering approach. As we can see from the vertical lines, characters appear in consecutive indexed books.

available weights for DASS, a YOLOX-based model for character and face detection. We have also employed a transformer-based Magi model [5] and a zero-shot transformer-based model for open-vocabulary detection called GroundingDino [4]. The results, of the detection benchmarks, are reported in Table 3. However, as some models are only trained for detecting a limited number of classes (DASS only detects faces and characters, while Magi does not detect faces). As a fair comparison, in Table 4 we provide the detection metrics calculated only based on the detectable classes.

Table 3: Results on **Detection task**, values reported are the mean Average Precision (mAP) over the four classes: Panel, Character, Text, Face.

| Models | DCM | eBD | comics | Pop | avg |
|---|---|---|---|---|---|
| G.Dino[4] | 49,5 | 37,7 | 49,2 | 48,9 | 48,7 |
| R-CNN[4] | 63,7 | 36,8 | **71,1** | 52,2 | **62,7** |
| YOLO[4] | **65,2** | **55,9** | 64,9 | 54,7 | 61,3 |
| DASS[2] | 41,3 | 21,8 | 34,2 | 20,9 | 30,4 |
| Magi[3] | 63,2 | 42,4 | 57,2 | **62,1** | 58,9 |

Table 4: Results on **Detection task**, values reported are the mean Average Precision (mAP) over the predictable classes (DASS only detects Character and Face, and MAGI does not detect Face.)

| Models | DCM | eBD | comics | Pop | avg |
|---|---|---|---|---|---|
| G.Dino[4] | 49,5 | 37,7 | 49,2 | 48,9 | 48,7 |
| R-CNN[4] | 63,7 | 36,8 | 71,1 | 52,2 | 62,7 |
| YOLO[4] | 65,2 | 55,9 | 64,9 | 54,7 | 61,3 |
| DASS[2] | 82,5 | 43,5 | 68,4 | 41,8 | 60,8 |
| Magi[3] | **84,3** | **56,5** | **76,3** | **82,9** | **78,6** |

## 4.2 Speaker identification

Regarding speaker identification, we provide two baseline results: using Magi or connecting the textbox with the closest character, within the panel. In Table 5 are reported the results for the two baselines, with metric $Recall$@#text as previously proposed [3].

## 4.3 Character Naming and Dialog generation

For the tasks of character naming and dialog generation, as detailed in the paper, we introduce the Hybrid Dialog Score (*HDS*), which combines the ANLS metric for assessing the accuracy of character

Table 5: Results on **Speaker identification**, the task of connecting the textbox with the speaker character within the page. Values reported are the $R@\#$text calculated on every page and averaged over the *CoMix* datasets.

| Models | DCM | eBD | comics | Pop | avg |
|---|---|---|---|---|---|
| closest | **42,0** | **67,1** | **36,1** | 37,3 | **38,4** |
| Magi | 13,2 | 13,1 | 15,0 | **57,2** | 27,9 |

names and the edit distance for evaluating the similarity between generated and ground truth dialog transcriptions. The methodology for computing the *HDS* metric is elaborated in Algorithm 2.

---

**Algorithm 2** Hybrid Dialog Score

---

1: **procedure** EVALUATETRANSCRIPTION($model\_output, ground\_truth$)
2:     $matches \leftarrow$ find optimal matches($model\_output, ground\_truth$)
3:     $tot\_ed, char\_name\_score \leftarrow 0, 0, 0$
4:     **for** each $(mo, gt)$ in $matches$ **do**
5:         $edit\_dist \leftarrow$ calculate edit distance($mo.text, gt.text$)
6:         $tot\_ed \leftarrow tot\_ed + edit\_dist$ / len($gt.text$)
7:         $anls\_score \leftarrow$ calculate ANLS($mo.name, gt.name$)
8:         $char\_name\_score \leftarrow char\_name\_score + anls\_score$
9:     **end for**
10:     $tot\_ed \leftarrow 1 - tot\_ed$
11:     $char\_name\_score \leftarrow char\_name\_score$ / len($matches$)
12:     **return** $tot\_ed, char\_name\_score$
13: **end procedure**

---

Specifically, the results for Character Naming are summarized in Table 6. Notably, the accuracy for the character naming task within the comic-style collection of the *CoMix* dataset is particularly low for the Magi system. This is primarily because Magi does not explicitly solve the character naming task but rather assigns names such as "Char n", where $n$ is an incremental identifier for character clusters. In contrast, within the PopManga collection, Magi exhibits enhanced performance, even outperforming GPT-4. This improvement can be attributed to the simpler task of analyzing single-page manga-style comics, where dialog is less frequent and character interactions are less complex compared to traditional comics. Consequently, the predominance of unknown characters in manga-style comics allows Magi to achieve higher average ANLS scores than GPT-4.

Table 6: **Results on Character Naming and Dialog Generation.** The metrics reported are ANLS for Character Naming and minimum edit distance for Dialog Generation. Both metrics are calculated on every page and averaged over the *CoMix* datasets.

**Character Naming**. The ANLS metric measures the similarity of two strings, considering it wrong when more than 50% do not agree.

| Models | DCM | eBD | comics | Pop | avg |
|---|---|---|---|---|---|
| Magi | 9,0 | 7,0 | 8,0 | 45,0 | 19,76 |
| GPT-4 | 54,0 | 37,0 | 58,0 | 28,0 | 47,11 |

**Dialog Generation**. The minimum edit distance metric is calculated for every possible match between ground truth and predicted dialogs.

| Models | DCM | eBD | comics | Pop | avg |
|---|---|---|---|---|---|
| Magi | 54,0 | 42,0 | 42,0 | 43,0 | 43,61 |
| GPT-4 | 93,0 | 94,0 | 93,0 | 89,0 | 93,14 |

However, the dialog transcriptions generated by Magi are of inferior quality compared to those produced by GPT-4, which achieved an impressive average *HDS* of 93.14%. This disparity highlights the strengths and limitations of the respective systems in handling complex narrative elements within the *CoMix* dataset.

# 5 Additional Ethical Considerations

Alongside the copyright and consent information and the initial biases analysis featured in the main paper, we include information on automatic semantic harmful content analysis using cutting-edge Multimodal LLMs.

## 5.1 Semantic Harmful Content

The dataset has undergone a thorough inspection by the authors to ensure it is free from NSFW and offensive content. Throughout the inspection process, no content from the existing datasets (DCM, eBDtheque, PopManga) or the newly included American golden-age comics was deemed NSFW.

Regarding offensive content, it is acknowledged that comics from the 1950s, both American and European, occasionally used terms that were mildly offensive towards minorities and wartime adversaries (such as Japanese or German soldiers), which were culturally tolerated during that period. However, following the implementation of the Comics Code by the Comics Code Authority[6], content depicting offensive themes or inspiring violence was strictly prohibited. Consequently, our selection is confined exclusively to comics published post 1954, ensuring all included works are compliant with the Comics Code. This careful curation supports the use of the dataset in diverse research and educational settings without risking exposure to inappropriate material.

Moreover, performing a full manual analysis of the semantic content and possible biases present in the *CoMix* dataset is complicated, and prone to subjective biases of human evaluators. We have instead opted to perform an automatic analysis of the textual content at the panel level, using the Llama3-80B model. We provided the model with detailed descriptions of different semantic classes and classified each panel accordingly (see the classes in Table 7).

The model was provided with detailed descriptions of different semantic classes, as shown below:

---

**Model Prompt**

You are an image classifier trained to identify the offensiveness of comic balloon content. You will be provided with the extracted content of a panel, in a single row, which represents the concatenation of multiple balloons, following the reading order. Based on that, you must choose a number from -1 to 5 as one of the following:
<TABLE 7>
Examine the following text and determine the offensiveness score (-1, 0, 1, ..., 5). Respond only with the identified style, without any explanation.
<text>

---

Table 7: Classes provided to the model. The critical panels are the ones classified as 3, 4, and 5.

| Code | Class | Short Description |
|---|---|---|
| -1 | Empty | The text is either empty or provides content which meaning is 'empty text'. |
| 0 | Neutral | Completely neutral content with no offensiveness. |
| 1 | Informal | Casual content, maybe mild slang but not offensive. |
| 2 | Humorous/Sarcastic | Humorous content, unlikely to offend most people. |
| 3 | Sexual/Angry/Violent | Sexual, angry, or violent content, potentially inappropriate for some but not offensive. |
| 4 | Offensive | Insulting content, but not racist or sexist. Just clearly offensive. |
| 5 | Highly Offensive/Racist/Sexist | Extremely racist, sexist, or offensive. |

---

[6] https://cbldf.org/comics-code-history-the-seal-of-approval

Table 8: Semantic harmful content analysis.

| code | class | percentage % |
|------|-------|--------------|
| -1 | Empty | 23.9 % |
| 0 | Neutral | 13.3 % |
| 1 | Informal | 14.5 % |
| 2 | Humorous/Sarcastic | 34.2 (34.7) % |
| 3 | Sexual/Angry/Violent | 12.7 (13.1) % |
| 4 | Offensive | 1.6 ( 1.2 ) % |
| 5 | Highly Offensive/Racist/Sexist | 0.8 (0.3) % |

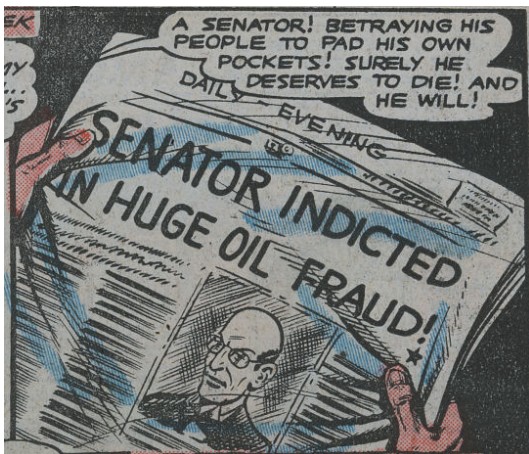

Figure 8: Example of a "Highly Offensive/Racist/Sexist" image (based on textual information) that, instead, belongs to "violent" or "incitement to violence" class.

From the initial analysis, we were able to obtain the results shown in Table 8.

We subsequently manually checked 370 panels to verify these results (from classes 3,4,5). We found that many of these are actually less (or not) offensive than the model classifications. An example of a panel being classified as "Highly Offensive/Racist/Sexist" is provided in Figure 8.

We updated the percentage in Table 8 considering these manual verifications. These statistics are meant to be indicative of the semantic content of our dataset, while we consider that a full, detailed study is out of the context of this manuscript.