# OpenReview forum: "CoMix: A Comprehensive Benchmark for Multi-Task Comic Understanding"
_NeurIPS.cc/2024/Datasets_and_Benchmarks_Track — NeurIPS 2024 Track Datasets and Benchmarks Poster_

### Official Review · Reviewer_7Mtd · 2024-07-24
**Good dataset with ethical concerns**

**Rating:** 7
**Confidence:** 5
**Correctness:** Yes
**Clarity:** Yes

**Review:**

Pros:

- The quality of the work is exemplary, demonstrating a comprehensive approach to addressing the multi-faceted challenges in comic analysis.

- The paper is well-structured and clearly written.

- This work has advanced the field of comic understanding significantly.

Cons:

- One of the main concerns with the CoMix benchmark is the relatively small volume of data, which could potentially limit the dataset's applicability in training.

- The dataset raises significant copyright concerns.

- The paper highlights issues of previous works, such as the availability of links, yet it relies on the same way for sharing data, which might eventually lead to the same problems. More sustainable solutions for data hosting and access are needed to ensure that the dataset remains accessible to the research community.


- The ethical considerations associated with comic content require more attention. While the paper briefly mentions gender representation in images, it lacks a thorough examination of potential biases in textual content within the comics.

**Strengths:**

See Review.

**Additional Feedback:**

NO

**Documentation:**

Yes

**Ethics:**

worry about copyright and bias problems

**Limitations:**

See Review.

**Opportunities For Improvement:**

See Review.

**Relation To Prior Work:**

Yes

**Summary And Contributions:**

The paper introduces CoMix, a novel benchmark designed to evaluate the multi-task capabilities of models in comic analysis. Unlike existing datasets that focus on isolated tasks and often feature a single comic style, CoMix integrates three existing datasets and introduces a new American comic-style dataset to ensure diversity. It offers extensive annotations for a range of tasks including object detection, speaker identification, character re-identification, reading order, character naming, and dialogue generation. The benchmark sets new standards with metrics such as Hybrid Dialog Score for evaluating higher-level tasks. Baseline results show significant performance gaps between current models and human performance, highlighting opportunities for advancements in comic understanding. The dataset, annotations, and evaluation server are publicly available, encouraging further research and development in this field.

---

> ### Author Rebuttal · Authors · 2024-08-14
>
> Dear Reviewer 7Mtd,
>
> Thank you for your constructive feedback and for highlighting several areas needing clarification and improvement in our submission.
>
> 1. Data Volume: We acknowledge the concern regarding the relatively small volume of our dataset. Rather than viewing this as a limitation, we see it as an essential feature of our specialized approach, which leads us to build a targeted benchmark dataset. Multimodal models are typically trained with large volumes of data, ranging from real images to infographics. However, despite the vast and costly amounts of training data, these models often fall short in analyzing the compositionality of comic and manga pages, and thus perform suboptimally on tasks that are simple yet complex in this specific context. To address this gap, we have deliberately curated our dataset to enhance the capability of models specifically tailored for the unique challenges of comics and manga. This strategic focus sets the groundwork for future advancements in training data and model development specifically for comic and manga analysis.
>
> 2. Copyright and Hosting: We have addressed your concerns regarding copyright and data hosting in the Rebuttal, ensuring all datasets are used within legal frameworks. Additionally, we are working towards hosting collaborations to enhance data accessibility and stability.
>
> 3. Textual Content Biases: Your observation regarding the need to examine biases within the textual content is particularly valuable. In response, we will introduce a dedicated subsection titled "Discrimination, Biases, and Fairness" within the "Ethical Considerations" section of our manuscript. This subsection will focus on the analysis of textual content, including dialogue and narration, to identify and address potential biases. By employing text classification techniques, we aim to detect and mitigate issues related to discriminatory language or biased representations, ensuring a more fair and inclusive dataset.

---

> > ### Author Rebuttal · Authors · 2024-08-31
> >
> > Dear Reviewer 7Mtd,
> > We have uploaded the revised rebuttal in PDF format in response to comments on ethical concerns. Please take a moment to review the updates at your earliest convenience. Thank you for your valuable feedback.

---

### Official Review · Reviewer_CRQ4 · 2024-07-27
**Useful benchmark for specialized visual understanding**

**Rating:** 6
**Confidence:** 3
**Correctness:** Good
**Clarity:** This paper is well written in general.

**Review:**

The reviewer acknowledges the authors effort in constructing this benchmark, given the working load for annotation and benchmarking. Apart from the dataset, they also propose high-level tasks metrics, i.e., Hybrid Dialog Score for character naming and dialog generation. However, as a potential flaw, comic understanding is quite a narrow field, and the authors do not provide the technical gap wrt the general domain.

**Strengths:**

Pros:
1.	Comprehensive comic content types and annotation;
2.	Benchmarking multiple types of SOTA algorithm.

**Additional Feedback:**

NA

**Documentation:**

Good

**Ethics:**

No concerns in this perspective.

**Opportunities For Improvement:**

Cons:
1.	The significance of this dataset can be questioned. How is comic understanding different from general visual understanding? What is the algorithmic gap (apart from data gap)?

**Relation To Prior Work:**

The prior works are properly discussed.

**Summary And Contributions:**

In this submission, the authors propose a novel comsic understanding benchmark, which supports many tasks such as Object detection, Speaker identification, Re-ID, etc. The annotation and task types are greatly extended compared to previous datasets.

---

> ### Author Rebuttal · Authors · 2024-08-14
>
> Dear Reviewer CRQ4,
>
> Thank you for your excellent question regarding how comic understanding differs from general vision understanding. We appreciate this opportunity to clarify the unique challenges presented by comics, which involve not just data differences but also structural and interpretative complexities inherent to comic formats. Comics necessitate understanding the sequential and spatial arrangement of panels, as well as the implicit content between panels (gutters) akin to scene transitions in films but confined to a single page.
>
> Your question has prompted us to emphasize these aspects more clearly in our revised manuscript, delineating the algorithmic challenges specific to comics apart from the general vision tasks. This differentiation is crucial for advancing domain-specific AI technologies.
>
> We are grateful for your insightful query, which has significantly enhanced the clarity of our discussion on the unique challenges in comic book analysis.

---

> > ### Author Rebuttal · Authors · 2024-08-31
> >
> > Dear Reviewer CRQ4,
> > We have uploaded the revised rebuttal in PDF format in response to comments on ethical concerns. Please take a moment to review the updates at your earliest convenience. Thank you for your valuable feedback.

---

### Official Review · Reviewer_voB9 · 2024-08-26
**The dataset is comprehensive**

**Rating:** 6
**Confidence:** 4
**Correctness:** The claims are correct.
**Clarity:** The paper is written clearly and easy…

**Review:**

- The introduced dataset contains a large amount of comic images with sufficient diversity.

- The annotations are comprehensive.

- The dataset is a good benchmark for evaluating multi-modal understanding ability of various models.

**Strengths:**

- The dataset is presenting challenging and interesting testing problems for multi-modal models.
- The dataset is comprehensive in terms of the annotations it has.

**Additional Feedback:**

See the comments above

**Documentation:**

The paper provides sufficient details.

**Limitations:**

- The scope of this dataset is kind of narrow.

**Opportunities For Improvement:**

- Comic understanding is kind of a narrow domain. It is unclear whether the model can transfer its good comic understanding abilities to other realistic world understanding problems (like understanding realistic images/videos).

**Relation To Prior Work:**

The prior works are well discussed.

**Summary And Contributions:**

This work introduces a new dataset consisting of comprehensively annotated comic images and texts. The dataset will be a good test benchmark for evaluating the comic understanding ability and multi-modal understanding ability of various multi-modal models.

---

> ### Author Rebuttal · Authors · 2024-08-31
>
> Dear Reviewer voB9,
> We have answered similar comments to Reviewer CRQ4, please check that discussion.
> Moreover, we have uploaded the revised rebuttal, as a PDF. Please take a moment to review the updates at your earliest convenience. Thank you for your valuable feedback.

---

### Author Rebuttal · Authors · 2024-08-14

# Ethical Considerations

## Copyright and Consent

Our dataset, *CoMix*, incorporates comics from various sources, including American comics from the Digital Comic Museum, manga from PopManga, and French Bande Dessinée from eBDtheque.

The Digital Comic Museum assets are confirmed to be in the public domain, either published without copyright or with expired copyrights, ensuring they can be freely used for research [1].

The PopManga source images are freely and publicly available on their site "Manga Plus by Shueisha," with official permissions from their copyright owners.

The eBDtheque’s database is publicly available data, cleared by their copyright holders for non-commercial research use, aligning with our research objectives [2].

## Data Quality and Representativeness

The *CoMix* benchmark is designed to evaluate models across diverse comic styles, including American, Japanese, and European works, which cover the main comics production hubs.

Our dataset is mainly focused on out-of-copyright works, especially for the American comics part. This means that most works are dated from the 1950s, and it is therefore possible that certain social biases and stereotypes of that era are inherent to the stories, as with any dataset based on historical documents. To this extent, we will take the following steps. First, we will acknowledge this possibility and include in the final version of this manuscript more statistics about factors like the gender of the main characters, the countries mentioned, etc. In parallel, we are grateful for the reviewer’s suggestion to introduce mechanisms such as periodic reviews of model performance across different populations to monitor how models perform on our benchmark data in terms of addressing such biases.

Our aim is to enhance global applicability and minimize cultural over-representation by including, in future iterations, also underrepresented styles such as Webtoons and Manhwa and incorporating other languages.

To ensure fairness and minimize biases in our dataset and models, we have implemented several strategies:
- **Diverse Dataset Composition**: We ensure our dataset encompasses a wide range of comic styles and origins to provide a balanced view that diminishes cultural bias.
- **Bias Detection and Analysis**: We will include in the final version of the manuscript more statistics focusing on gender, ethnicity, style, and language representation. Future updates will incorporate more detailed annotations and classification for a comprehensive analysis.

By implementing these strategies, we strive to set a new standard for ethical research in comic analysis [3], ensuring that our work is both innovative and respectful of diverse cultural norms.

---

**References:**

[1] Digital Comic Museum, Usage terms, 2024. [Link](https://digitalcomicmuseum.com/forum/index.php/topic,2759.0.html).

[2] eBDtheque website, 2024. [Link](https://ebdtheque.univ-lr.fr).

[3] European Journal of American Studies, The Mutant Problem: X-Men, Confirmation Bias, and the Methodology of Comics Identity, 2015. [Link](https://journals.openedition.org/ejas/10890).

---

> ### Author Rebuttal · Authors · 2024-08-31
>
> Dear Reviewers,
> We have uploaded the revised rebuttal, as a PDF, in response to your comments. Please take a moment to review the updates at your earliest convenience. Thank you for your valuable feedback.

---

### Decision · Program_Chairs · 2024-09-26

**Decision:**

Accept (Poster)

**Comment:**

This work introduces a new dataset with extensively annotated comic images and texts, designed to serve as a benchmark for evaluating comic understanding and multi-modal capabilities of various models.

Reviewers concur that the dataset is comprehensive and valuable for comic understanding research. However, they have raised significant concerns regarding potential bias and copyright issues. The authors have committed to addressing these concerns by providing explicit explanations of copyright and ethical considerations in the final version. They are strongly encouraged to ensure the dataset is used exclusively for research purposes.